# Effects of 10-Week Online Moderate- to High-Intensity Interval Training on Body Composition, and Aerobic and Anaerobic Performance during the COVID-19 Lockdown

**DOI:** 10.3390/healthcare12010037

**Published:** 2023-12-23

**Authors:** Lorena Rodríguez-García, Halil Ibrahim Ceylan, Rui Miguel Silva, Ana Filipa Silva, Amelia Guadalupe-Grau, Antonio Liñán-González

**Affiliations:** 1Department of Physical Activity and Sport Sciences, Pontifical University of Comillas, 07013 Palma, Spain; lrodriguez@cesag.org; 2SER Research Group, Pontifical University of Comillas, 07013 Palma, Spain; 3Physical Education and Sports Teaching Department, Kazim Karabekir Faculty of Education, Ataturk University, 25240 Erzurum, Turkey; halil.ibrahimceylan60@gmail.com; 4Escola Superior Desporto e Lazer, Instituto Politécnico de Viana do Castelo, Rua Escola Industrial e Comercial de Nun’Álvares, 4900-347 Viana do Castelo, Portugal; 5Delegação da Covilhã, Instituto de Telecomunicações, 1049-001 Lisboa, Portugal; 6GENUD Toledo Research Group, Universidad Castilla-La Mancha, 45002 Toledo, Spain; amelia.guadalupe@uclm.es; 7CIBER of Frailty and Healthy Aging (CIBERFES), Instituto de Salud Carlos III (ISCIII), 28029 Madrid, Spain; 8Department of Nursing, Faculty of Health Sciences, Melilla Campus, University of Granada, 52005 Melilla, Spain; antoniolg@ugr.es

**Keywords:** performance, sedentary, women, anthropometric, interval training

## Abstract

The present study aimed to investigate the effects of a 10-week online high-intensity interval training (HIIT) program on body composition and aerobic and aerobic performance in physically sedentary women. A parallel, two-group, longitudinal (pre, post) design was used with physical tests performed before (preintervention) and after (postintervention) the 10-week intervention period. A total of forty-eight healthy and physically sedentary women (defined as an individual who lacks regular exercise or a structured fitness routine) were recruited to participate in this study. The participants were distributed in two groups: the experimental group (EG) with 24 women (mean ± SD: age 21.21 ± 2.15 years; weight: 61.16 ± 8.94 kg; height: 163.96 ± 4.87 cm; body mass index (BMI): 22.69 ± 2.49 kg/m^2^) and the control group (CG) with another 24 women (mean ± SD: age 20.50 ± 1.29 years; weight: 62.0 ± 6.65 kg; height: 163.92 ± 4.89 cm; body mass index: 23.04 ± 1.74 kg/m^2^). The EG performed an online HIIT program for 10 weeks, while the CG continued with their daily life routines. The repeated measures ANCOVA indicated a significant effect in the within-group analysis for weight (*p* = 0.001; *d* = −0.96) and for BMI (*p* = 0.001; *d* = 0.24), with a significant decrease in the experimental group (EG). The control group (CG) did not show any significant decrease in either body weight or BMI. Regarding the maximal oxygen uptake (VO2 max) values, the EG exhibited a significant improvement (*p* = 0.001; *d* = −1.07), whereas the CG did not demonstrate a significant improvement (*p* = 0.08; *d* = −0.37). The EG’s power output (W) (*p* = 0.001; *d* = −0.50) and power output standardized by body weight (W/kg) (*p* = 0.001; *d* = −0.96) were significantly improved. The CG did not show a significant improvement in either power output (W/kg) or power output. Lastly, the within-group analysis with load revealed that the EG significantly improved (*p* = 0.001; *d* = −0.50), while CG did not show a significant improvement in load (*p* = 0.10.; *d* = −0.10). The present study showed that 10 weeks of HIIT in an online environment during the COVID-19 lockdown significantly improved maximum oxygen consumption and caused weight loss and a significant decrease in body mass index in physically sedentary women. These results suggest that HIIT may be used as a time-efficient strategy to improve body composition and cardio-respiratory fitness in sedentary women.

## 1. Introduction

The COVID-19 pandemic has ushered in unprecedented challenges, significantly disrupting daily routines and prompting profound changes in lifestyle patterns worldwide [1]. Government-imposed lockdowns and restrictions on mobility have led to a surge in sedentary behaviors among individuals, exacerbating concerns about health and well-being [2]. Sedentary lifestyles are associated with various health issues, including increased risks of cardiovascular diseases, obesity, and mental health disorders [3]. The pandemic’s impact on physical activity underscores the critical need for innovative and accessible exercise solutions that can be implemented under restrictive conditions.

High-intensity interval training (HIIT) is estimated to be one of the most popular trends in fitness according to the American College of Sports Medicine (ACSM)’s Annual Fitness Trend Forecast [4]. HIIT is characterized by repeated short-term explosive-intensity anaerobic activities (≥85–90% maximal oxygen uptake (VO2 max) for health subjects or ≥80% VO2 max for clinical populations), interspersed by periods of passive or low-intensity exercise recovery, and it typically takes less than 30 min per training session [5,6]. HIIT has gained recognition as a popular alternative exercise approach compared to traditional moderate-intensity continuous exercise. This method aims to reduce the body fat percentage, lower the resting heart rate, and improve aerobic fitness in a shorter timeframe, particularly among sedentary individuals [7,8,9].

Similarly, HIIT provides significant enhancements in VO2 max. This contributes to improvements in both aerobic and anaerobic fitness within a brief period, surpassing the benefits offered by alternative training methods [10]. Therefore, previous studies suggested that HIIT could be a solution to improve health and reduce morbidity in the adult population [11,12]. This type of training is tolerable and acceptable to physically sedentary people, although at first, it generated a lot of controversy in people with little training due to the high intensity that this mode of training entails. However, in recent years, it has been prescribed to the elderly, young people, and adolescents [13]. Therefore, it must be taken into account that these HIIT programs can cause a greater feeling of fatigue due to a higher ratio of perceived exertion (RPE) in comparison with the classic continuous training of moderate intensity [14].

The context of the COVID-19 pandemic, which disrupted regular exercise routines, underscores the relevance of understanding the impact of online HIIT during periods of restricted mobility and social distancing. While previous literature has predominantly delivered HIIT in person, rather than conducting it online [7], the present study explored the novel approach of online delivery and considered the unique challenges and opportunities it presents. Therefore, the present study may contribute valuable insights into promoting accessible and effective exercise strategies, particularly during times of restricted mobility and social isolation.

Understanding how an online HIIT approach can impact individuals during a period of restricted mobility and social distancing is highly relevant. For the above reasons, the purpose of this study was to investigate the effects of a 10-week online HIIT intervention on body composition and aerobic performance variables in healthy, physically sedentary women.

## 2. Materials and Methods

### 2.1. Experimental Approach to the Problem

A parallel, two-group, longitudinal (pre, post) design was used with physical tests performed before (preintervention) and after (postintervention) the 10-week intervention period. The participants were assigned and matched into two groups, an experimental group (EG) and a control group (CG). The participants from the CG were asked to maintain their daily life routines, while those from the EG were introduced to a one-hour HIIT familiarization session before the start of the intervention. Regarding the day of training, it was preceded by 48 h of absence of high effort activities. The study was conducted between January and March 2021. (See Table 1, for more information).

### 2.2. Participants

A total of forty-eight physically sedentary women were recruited to participate in this study. There were two groups: the EG with twenty-four women (*n* = 24; age: 21.21 ± 2.15 years; weight: 61.16 ± 8.94 kg; height: 163.96 ± 4.87 cm; body mass index: 22.69 ± 2.49 kg/m^2^) and the CG with another twenty-four women (*n* = 24; age: 20.50 ± 1.29 years; weight: 62.0 ± 6.65 kg; height: 163.92 ± 4.89 cm; body mass index: 23.04 ± 1.74 kg/m^2^). The randomization sequence was generated electronically (https://www.randomizer.org, accessed on 5 January 2021) and was concealed until the interventions were assigned.

Moreover, a priori sample size calculation was performed using a free online tool, G*Power (www.gpower.hhu.de, accessed on 3 January 2021), with a power level of 95% and an α level of 0.05 and based on previous and similar studies [15]; it revealed that a sample size of >36 would be sufficient for conducting a randomized controlled trial. The sample group for this study was selected using the criterion sampling method, which falls under the category of purposive sampling techniques [16]. The inclusion criteria for the participants in this study were (i) being physically sedentary (not engaging in 150 min per week of moderate-intensity exercise or 75 min per week of vigorous-intensity exercise or a combination equivalent to these two different intensities); (ii) not presenting any injuries; (iii) giving their consent; and (iv) participated in at least 90% of the training sessions during the intervention.

Finally, the participants obtained information on the main objectives of the research and signed the informed consent form. All the participants in this research were treated according to the guidelines of the American Psychological Association (APA), and therefore the anonymity of the participants’ responses was guaranteed. The study was carried out in accordance with the ethical principles of the Declaration of Helsinki for research in humans and was approved by the Research Ethics Committee of the Pontifical University of Comillas (2021/85).

### 2.3. Procedures

The forty-eight physically sedentary women enrolled in this study visited the laboratory twice, both for pre-test and post-test assessments, between 9:30 a.m. and 5:00 p.m. During the pre- and post-assessments, the participants underwent a sub-maximal incremental fitness test on a stationary ergometer. Adhering to the guidelines outlined by the American College of Sports Medicine (2018), measures were taken to ensure the safety of the physically sedentary women. Body mass, without shoes, was measured utilizing a bioelectrical impedance analysis (BIA) device (Tanita BC-730) accurate to the nearest 0.1 kg. Height was measured using a stadiometer (Type SECA 225, Hamburg, Germany) accurate to the nearest 0.1 cm. Body mass index was computed as mass (in kilograms) divided by the square of height (in meters).

All participants performed a sub-maximal incremental fitness test on a cycle ergometer (Viasprint 150 P cycle-ergometer) connected to a Jaeger Master Screen gas analyzer. Determination of the ventilatory anaerobic threshold (VAT) was based on the respiratory gas exchange method (RER) (RER = CO_2_ production/O_2_ consumption), which detected the VAT at the point at which RER exceeds the cut-off value of 1.0 [17], as well as the obtained maximal oxygen uptake. First, an RS800CX Polar monitor (Polar Electro, Helsinki, Finland) was used to monitor and record the maximal heart rate (HRmax) during the pre- and post-assessments. In this sense, the protocol consisted of a submaximal incremental test with a fixed cadence of 60 revolutions per minute (rpm). The warm-up started at 0 W and the workload was increased by 10 W every min until min 5. The participant began the exercise phase pedaling at 50 W, and the workload was increased to 25 W every two min. After each increase, the workload remained stable for the next 2 min. The submaximal test ended once the VAT was reached. The highest power output (W) reached during the cycle ergometer test was recorded. Finally, the highest power output reached was normalized by the participant’s body weight (W/kg).

### 2.4. Training Program

The experimental group (EG) engaged in a 10-week online program, involving three sessions per week. The sessions comprised high-intensity interval training (HIIT) following the Tabata method, involving 4 min intervals with 8 intensive training blocks, each followed by 1 min of recovery [18]. The exercises involved body weights and functional movements, aligning with a protocol that demonstrated favorable outcomes in female university students [8]. Notably, the last session of each week focused on continuous running, as outlined in Table 2.

In the first month, they performed 20 s intervals of intense exercise with 10 s of recovery, repeating this for 4 min. In the second month, they increased the technical difficulty of the exercises and increased the volume of the sessions until they achieved 20 min of Tabata in the last week of the intervention.

The HIIT sessions were carried out twice a week and their intensity was monitored daily using the Borg’s Scale (6–20) of the rate of perceived exertion (RPE) [19]. Each participant was asked, “How did you perceive exertion during the exercise execution?”. The first four weeks had an RPE of 12–13, indicating a somewhat hard level. The following four weeks were at an RPE of 15–16, signifying a hard level. In the last two weeks, the RPE was 18–19, representing an extremely hard level. However, the HR was not monitored during the intervention.

The participants were instructed to perform the exercise at their maximum exertion while maintaining the correct technique. The structure of the session consisted of a 10–15 min warm-up, a 35 min main part, and a 10 min cooldown. On the third day of each training week, the participants completed a moderate-intensity session outdoors, combining continuous running with walking.

Technical progressions in exercise complexity were systematically implemented throughout the experimental period. The progression for push-ups involved starting with hands against the wall, followed by positioning knees on the floor, and advancing to performing the exercise without resting the knees. Planks transitioned from forearms on the floor to arms fully extended. Similarly, jumping jacks initially began as static and progressed to dynamic, while squats advanced from using a box to incorporating a jump. This systematic approach extended to all exercises. During the first two weeks of the intervention, the participants followed an 8 min Tabata protocol, adhering to 20 s of intense exercise with 10 s of recovery, repeated for 4 min. Subsequently, we systematically increased the Tabata duration in the following weeks. More precisely, we implemented a progressive overload strategy by incorporating an additional minute every two weeks. By the conclusion of the 10-week intervention, the Tabata duration had reached a total of 20 min.

Due to the COVID-19 confinement regulations mandated by the Spanish government, the training program had to be conducted inside participants’ homes, where the absence of equipment led to the selection of calisthenics exercises.

### 2.5. Statistical Procedures

The data were analyzed using Statistica software (version 10.0; Statsoft, Inc., Tulsa, OK, USA). Finally, the significance level was set at *p* < 0.05. Normal distribution and homogeneity tests (Kolmogorov–Smirnov and Levene’s, respectively) were conducted on all metrics. A paired sample *t*-test was used for determining differences as a repeated measures analysis (pre and post). Cohen’s d was used as the effect size indicator. To interpret the magnitude of the effect size, we adopted the following criteria: d ≤ 0.20, small; d ≤ 0.50, medium; and d ≤ 0.80, large. To elucidate the between-group differences, an ANCOVA test was performed using the pre-test as a covariate and the pre and post times as factors. To interpret the magnitude of the effect size of ANCOVA, we adopted the following criteria: ηp^2^ = 0.02, small; ηp^2^ = 0.06, medium; and ηp^2^ = 0.14, large.

## 3. Results

Descriptive statistics were calculated for each variable (Table 3). No significant baseline between-group differences were recorded for all measurements (*p* > 0.05, d = 0.04–0.31). Moreover, the adherence rate to the online HIIT program in the experimental group was 96.3% ± 2.2%, and no withdrawals were reported.

Significant group*time interactions were observed through repeated measures ANCOVA for weight (*p* = 0.001; ηp^2^ = 0.97), body mass index (*p* = 0.001; ηp^2^ = 0.35), VO2 max (*p* = 0.001; ηp^2^ = 0.42), power output (W/kg) (*p* = 0.001; ηp^2^ = 0.49), and power output (W) (*p* = 0.001; ηp^2^ = 0.87). In the EG, weight significantly decreased (*p* = 0.001; d = −0.96), while the CG did not show a significant change (*p* = 0.15; d = −0.11) in body mass index. In terms of VO2 max, the EG demonstrated significant improvement (*p* = 0.001; d = −1.07), while the CG did not exhibit a significant change (*p* = 0.08; d = −0.37). A separate analysis of power output (W) revealed a significant improvement in the EG (*p* = 0.01; d = −0.96), while the CG did not show a significant change in power output (W/kg) (*p* = 0.20; d = −0.11). Lastly, the analysis of the power output (W) showed there was significant improvement in the EG (*p* = 0.001; d = −0.50), whereas the CG did not show a significant change (*p* = 0.10; d = −0.10). Refer to Figure 1 and Figure 2 for a graphical representation.

## 4. Discussion

The purpose of this study was to investigate the impact of a 10-week online high-intensity interval training (HIIT) program on body composition and aerobic performance during the COVID-19 lockdown. The main evidence of the present study revealed that significant reductions in body weight were observed after the 10-week HIIT program compared to the CG. Moreover, there were significant improvements in aerobic fitness after the HIIT program compared to the CG.

The results of the present study showed that online HIIT can be an effective stimulus for weight loss and decreasing BMI. Significant weight loss (effect size: 0.41, moderate, −3.22%) and a decrease in body mass index (effect size: 0.24, moderate, −2.72%) compared to the control group. This aligns with recent studies reporting body composition improvements within 8 weeks [8] and 15 weeks [20] of HIIT in sedentary young females. The effectiveness of online HIIT in promoting weight loss and reducing BMI in our study may be attributed to increased post-exercise energy expenditure. HIIT is known to induce excess post-exercise oxygen consumption (EPOC), leading to elevated oxygen consumption and calorie expenditure, contributing to a negative energy balance and, consequently, weight loss [21]. Contrary to our findings, previous studies [22,23,24,25] reported no significant changes in body weight, BMI, and body fat percentage after HIIT in sedentary young females. Differences in participant characteristics and exercise program content (duration, frequency, intensity, volume, and progression) could contribute to these varying results [25,26].

A 10-week online HIIT intervention led to significant improvements in VO2 max, power output (W/kg), and power output (W) when compared to the control group. This is in line with prior research, such as Lu et al. [8], that reported a substantial increase in VO2 max (+18.8%) following an 8-week HIIT in sedentary young females. Another study demonstrated noteworthy enhancements in VO2 max (approximately 22%) with cycling HIIT performed for 18–30 min per week over 12 weeks [22]. Moreover, evidence suggests that both low-volume (1 × 4 min treadmill running at 85–95% HRmax) and high-volume HIIT (4 × 4 min treadmill running at 85–95% HRmax), executed thrice weekly over 6 weeks, exhibit equal effectiveness in enhancing aerobic capacity (VO2 max) among sedentary young women [15]. A systematic review and meta-analysis on adults indicated that short-term HIIT (<12 weeks) and long-term HIIT (≥12 weeks) performed at least 3 times a week for 12 weeks can result in an increased VO2 max. Additionally, the increase in aerobic capacity was reported to be greater at longer training times [7]. Another meta-analysis demonstrated that HIIT ranging from 2 to 8 weeks lead to a substantial increase in VO2 max by 4.2–13.4% in healthy, sedentary/recreationally active adults, emphasizing its positive impact on aerobic capacity [27]. A recent systematic review underscores that HIIT, with a minimum of 4 weeks of exercise training (3 times per week, 18–30 min per session, 85–95% HRR), is effective for augmenting maximal aerobic capacity in women with a sedentary lifestyle [28].

Considering the above studies, it can be said that longer durations of HIIT are required for greater improvements in VO2 max. However, it was observed that HIIT exercises for 2–12 weeks improved cardiorespiratory fitness or aerobic performance. In addition, in our study, a 15.94% VO2 max increase was detected after 10 weeks of the HIIT protocol. Our results are in agreement with previous studies, which indicated that improvements of 9% to 13% were found in VO2 max following 8-week [29] and 12-week [30] HIIT protocols in sedentary individuals. Moreover, the observed improvement was attributed to the training intensity, emphasizing that aerobic exercise intensity plays a pivotal role in enhancing VO2 max. These findings underscore the significance of HIIT conducted at higher training intensities, specifically within the range of 90–95% HRmax, for achieving optimal training-induced enhancements in VO2 max, especially concerning age-related considerations [22,29]. The considerable enhancement in VO2 max observed can be ascribed to the exercise intensity implemented in the HIIT program. Additionally, the existing literature, which aligns with our study, highlights that individuals with a more sedentary lifestyle exhibit the most substantial VO2 max response [22,28].

Furthermore, previous studies have suggested several physiological mechanisms underlying the improvement in VO2 max after HIIT. For instance, in one study, it was suggested that the improvement in VO2 max or cardiorespiratory fitness after HIIT could be associated with several central adaptations, such as increases in systolic volume and cardiac output due to the increased cardiac contractility and oxygen availability. Also, it was asserted that peripheral changes such as increased muscle oxidative potential, increased skeletal muscle diffusion capacity, increased number and size of mitochondria, increased mitochondrial enzyme activity, arterial vasodilation, increased nitric oxide bioavailability, and reduced oxidative stress might be responsible for the improvement in VO2 max following the HIIT intervention [22,31]. Moreover, in another study, it was stated that HIIT-induced increases in VO2 max and improved maximal aerobic performance could partly be explained by specific oxidative adaptations in type II fibers [27].

This study has some limitations that should be considered. The first limitation of the study is related to the use of the cycle ergometer evaluation, which may not precisely reflect the intensities prescribed for running and jumping exercises in the online high-intensity interval training program. Secondly, it is important to acknowledge that relying solely on participants’ RPE without concurrent heart rate monitoring introduces a potential limitation. While RPE is a valuable subjective measure of effort, incorporating heart rate data would have provided an additional objective parameter to quantify the actual physiological intensity experienced by participants during each session. This absence of heart rate monitoring restricts the precision of intensity characterization and may limit the generalizability of our findings to a broader context where objective intensity measures are commonly employed. Thirdly, we did not control for other influencing factors that could influence our results. Furthermore, given that the sample consisted of sedentary individuals, the outcomes may have been impacted by the duration of exercise progressions. Therefore, future studies employing similar training protocols in sedentary populations might benefit from extending the intervention period.

## 5. Conclusions

The present study aimed to analyze the effects of a 10-week online HIIT program on body composition and aerobic performance during the COVID-19 lockdown. The main findings showed that the 10-week online HIIT intervention yielded significant improvements in maximum oxygen consumption and weight loss, and a noteworthy decrease in body mass index among sedentary young females. These findings underscore the efficacy of HIIT as a time-efficient strategy for enhancing both body composition and cardiorespiratory fitness in sedentary women, even under limited circumstances such as a confinement period.

## Figures and Tables

**Figure 1 healthcare-12-00037-f001:**
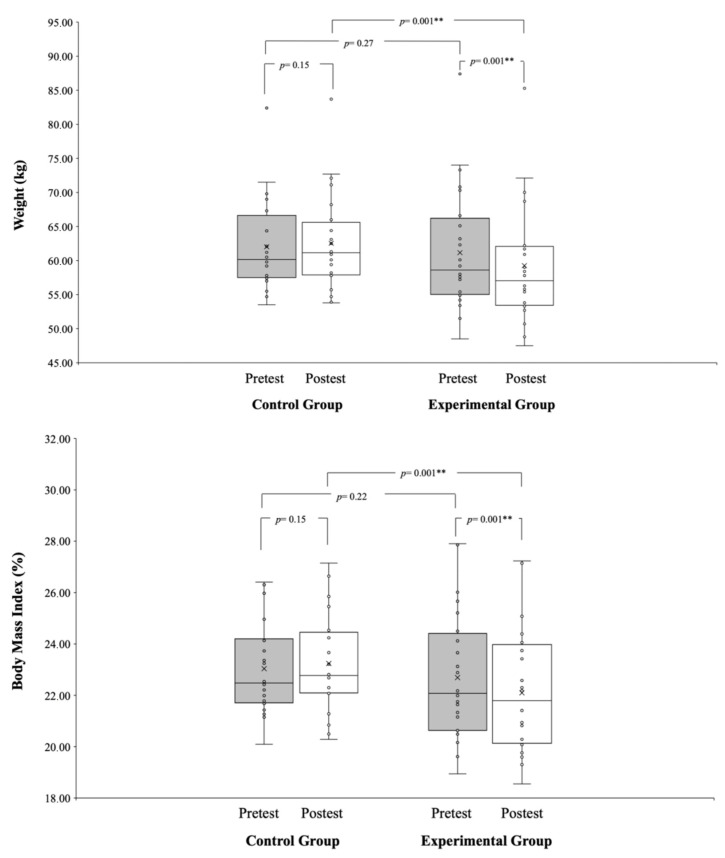
Pre- and post-tests anthropometrical measures (weight and body mass index) of CG and EG. ** denotes significance at *p* < 0.01.

**Figure 2 healthcare-12-00037-f002:**
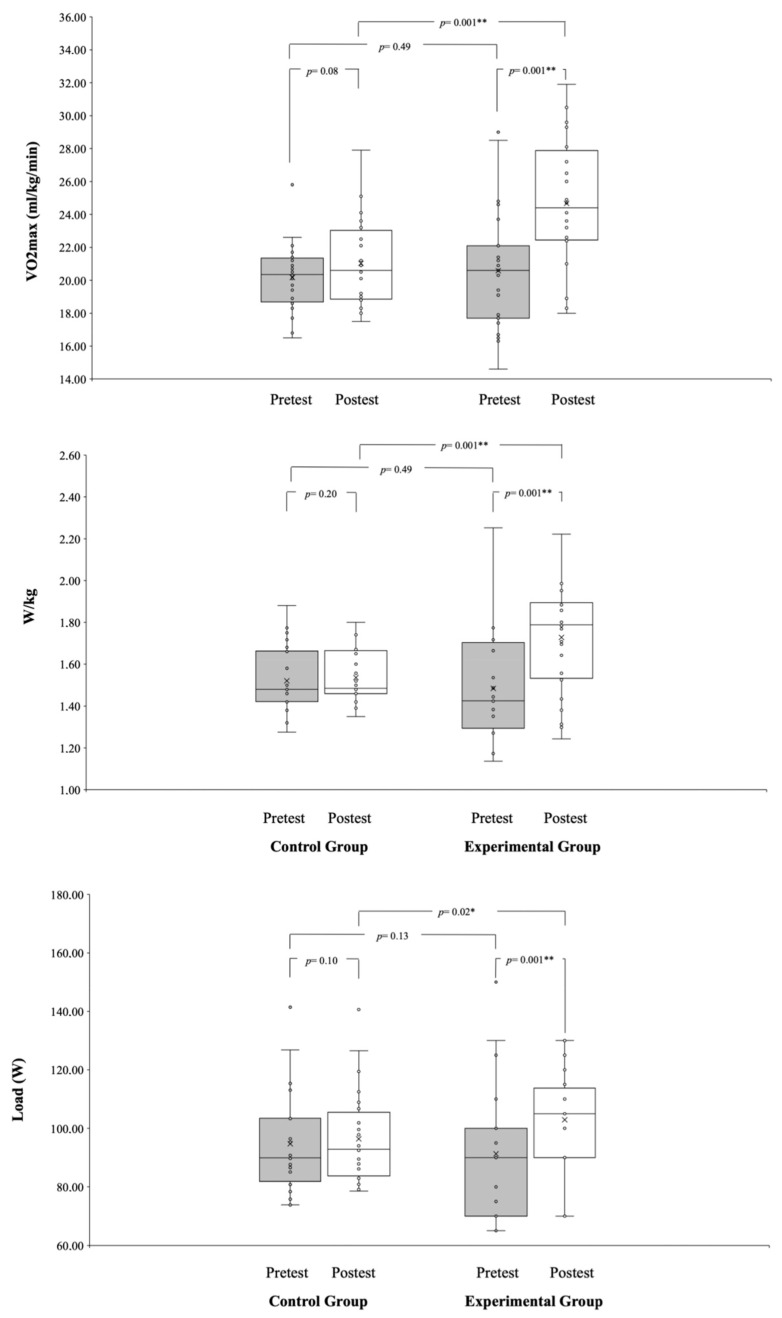
Pre- and post-tests performance variables (VO2 max, power output (W), and power output (W/kg)) of CG and EG. * denotes significance at *p* < 0.05. ** denotes significance at *p* < 0.01.

**Table 1 healthcare-12-00037-t001:** Timeline of this study.

	2021											
Months	January			February				March				
Week		1	2	3	4	5	6	7	8	9	10	
EG	Pre	High-intensity interval training										Post
CG	Pre	Maintain routines										Post

**Table 2 healthcare-12-00037-t002:** Example of HIIT and running session and work set progression during the HIIT program.

Example of Weekly Session			
	1st Session	2nd Session	3rd Session
Exercises	HIIT. Example 1	HIIT. Example 2	Running
1st	Push-ups	Jumping jacks	5 min of running at 65–75% HRmax
2nd	Mountain Climbing	Box squat	5 min of walking at 40–45% HRmax
3rd	Forearm plank	Bench step-ups	5 min of running at 65–75% HRmax
4th	Dips using a bench or chair	Deadlift w/elastic band	5 min of walking
Training progression during the HIIT program			
Week	Set/session	Recovery between sets (s)	
1st and 2nd	2	60	
3rd to 5th	3	60	
6th to 8th	4	60	
9th and 10th	4	45	

**Table 3 healthcare-12-00037-t003:** Performance variables before (pre-test) and after (post-test) the intervention period (mean ± SD).

Participants (*n* = 48)									
Control Group (*n* = 24)					Experimental Group (*n* = 24)				Differences Between Groups (ANCOVA Test)
	Pre-Test	Post-Test	%	RM *t*-Test (*p*)	Pre-Test	Post-Test	%	RM *t*-Test (*p*)	
Anthropometric measures									
Weight (kg)	62.00 ± 6.65	62.53 ± 7.07	0.75 ± 2.89	*p* = 0.15 *d* = −0.08	61.16 ± 8.94	59.26 ± 8.73	−3.22 ± 1.98	*p* = 0.001 ** *d* = 0.41	*F*(1,47) = 1244.36; *p* = 0.000; *ηp*^2^ = 0.97
Body Mass Index (%)	23.04 ± 1.74	23.24 ± 1.97	0.75 ± 2.89	*p* = 0.15 *d* = −0.11	22.69 ± 2.49	22.10 ± 2.47	−2.72 ± 1.98	*p* = 0.001 ** *d* = 0.24	*F*(1,47) = 685.01; *p* = 0.000; *ηp*^2^ = 0.35
Incremental test									
VO2 max (mL/kg/min)	20.17 ± 2.03	21.03 ± 2.59	3.37 ± 10.37	*p* = 0.08 *d* = −0.37	20.58 ± 3.65	24.67 ± 3.98	15.94 ± 12.08	*p* = 0.001 ** *d* = −1.07	*F*(1,47) = 32.38; *p* = 0.000; *ηp*^2^ = 0.42
Power output (W/kg)	1.52 ± 0.16	1.54 ± 0.12	1.18 ± 3.84	*p* = 0.20 *d* = −0.11	1.48 ± 0.26	1.73 ± 0.26	13.50 ± 11.72	*p* = 0.001 ** *d* = −0.96	*F*(1,47) = 43.21; *p* = 0.000; *ηp*^2^ = 0.49
Power output (W)	94.71 ± 17.32	96.48 ± 15.04	1.90 ± 5.21	*p* = 0.10 *d* = −0.10	91.26 ± 23.11	102.92 ± 23.86	10.67 ± 12.57	*p* = 0.001 ** *d* = −0.50	*F*(1,47) = 148.58; *p* = 0.000; *ηp*^2^ = 0.77

Note: %: percent change; RM: repeated measures ** denotes significance at *p* < 0.01.

## Data Availability

The raw data used to support this article’s conclusions will be made freely available.

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
