# Peer review of "Effects of 10-Week Online Moderate- to High-Intensity Interval Training on Body Composition, and Aerobic and Anaerobic Performance during the COVID-19 Lockdown"

_healthcare, 2023, doi:10.3390/healthcare12010037_

Round 1

Reviewer 1 Report (Previous Reviewer 2)

Comments and Suggestions for Authors

I have rereviewed the manuscript titled "Effects of 10-week interval training on body composition and performance variables in physically inactive women" for consideration for publication in Healthcare.

It is a very interesting study that will probably encourage other authors to carry out more studies. Congratulations to the authors.

Author Response

Reviewer 1

AUTHORS: Dear Reviewer, thank you so much for your great and detailed previous review. It helped us to improve the overall quality of our study.

Reviewer 2 Report (Previous Reviewer 3)

Comments and Suggestions for Authors

Please refer to the attached reviewers report. 

Author Response

Reviewer 2, Major and Minor revisions:

AUTHORS: Dear Reviewer, thank you so much for your great and detailed review. Below are all changes made, referencing each page and the lines where the changes were made. All changes in the manuscript regarding your suggestions are highlighted in YELLOW.

Major revisions

Title:

REVIEWER: Firstly, I still believe consideration needs to be given to the title to reflect the analysis and changes in power output between the groups, as I would not consider these measures of aerobic performance.

AUTHORS: P1, L1-4: Dear reviewer, thank you so much for your suggestion. We changed the title to “Effects of 10-Week Online Moderate- to High-Intensity Interval Training on Body Composition, Aerobic and Anaerobic Performance during the COVID-19 Lockdown”.

Methods:

REVIEWER: Second, although the description of the training program has been vastly improved, greater clarification and justification is warranted: (1) I still believe the authors need to consider clearly defining what they consider high-intensity exercise to be and how their program adhered to this definition, as it appears that some parts of the program were not prescribed at a high-intensity. For example (Lines163-165), the authors state that, on the third day of each training week, the participants completed a moderate-intensity session outdoors, combining continuous running with walking. If the authors in fact used a mixed-intensity approach to their training, then this should be reflected in the title and throughout the paper.

AUTHORS: Dear reviewer, thank you. We agree with you on this as we programmed a moderate-intensity session on each third session of each week. Given that, we have changed accordingly throughout. Furthermore, as coaches and researchers, we usually implement running-based HIIT programs that follow the different types and formats of HIIT, as in the incredible work of our colleagues Buchheit and Paul:

Buchheit, M., & Laursen, P. B. (2013a). High-intensity interval training, solutions to the programming puzzle: Part I: Cardiopulmonary emphasis. Sports Medicine, 43(5), 313–338. https://doi.org/10.1007/s40279-013-0029-x

Buchheit, M., & Laursen, P. B. (2013b). High-intensity interval training, solutions to the programming puzzle: Part II: Anaerobic energy, neuromuscular load and practical applications. Sports Medicine, 43(10), 927–954. https://doi.org/10.1007/s40279-013-0066-5

However, this intervention took place during the Covid-19 confinement period. Despite participants' concerted efforts to engage in at least one outdoor session weekly, there was initial hesitation towards outdoor exercise. Initially, we planned to conduct all three sessions in an outdoor setting, employing both short- and long-HIIT running-based formats. However, due to heightened fears surrounding the virus, exacerbated by government restrictions and threats faced by citizens on the streets during that period, we decided to shift our study to an online approach, retaining only one session conducted outdoors. This adjustment was made in response to the prevailing circumstances and concerns related to the pandemic.

REVIEWER: (2) Could the authors please add a reference to support their RPE definitions (Lines 171-174), as in my opinion, an RPE of 12-13 constitutes moderate-intensity exercise (Norton et al. 2010. Position statement on physical activity and exercise intensity terminology. Journal of Science and Medicine in Sports, 13, 496-502); which relates back to my previous point;

AUTHORS: P4-L160-166: Dear reviewer, thank you. We truly understand your concern regarding the interpretation of the Borg’s perceiver exertion 6-20 scale. However, as you can see in P4, L171-172, we clearly stated that “The first four weeks had an RPE of 12-13, indicating a somewhat hard level.”. In P4, L172-173 we stated “The following four weeks were at an RPE of 15-16, signifying a hard level”, and “In the last two weeks, the RPE was 18-19, representing an extremely hard level.”. We also told you in the first round, that we had to ensure HIIT progressions as the sample was comprised of sedentary individuals with no experience in any kind of exercise prescription. We have added the Borgs’ in-text citation and reference (it escaped us before).

Borg 6-20 RPE scale interpretation (see figure below):

The added reference: Borg, G. (1990). Psychophysical scaling with applications in physical work and the perception of exertion. Scandinavian Journal of Work, Environment & Health, 16(1), 55–58.

REVIEWER: (3) I also believe it is essential that the authors report the actual intensity that the participants achieved during each Tabata session, to clarify whether they were exercising at high-intensity. Simply stating, that the objective was to perform exercises at a maximal intensity (Lines 161-162) does not mean participants were exercising at a sufficient intensity for it to be considered HIIT. Given both heart rate and RPE were monitored, the intensity of each session or average intensity of the program should be reported.

AUTHORS: P4-L160-166: Dear reviewer, thank you for alerting us to this issue as it is crucial for the transparency of the study. I must express my previous error when making the first revisions round, as I wrongly stated that during the training sessions, the intensity was monitored using both RPE and HR. The intensity was only monitored using the RPE values. We have changed to the following: “The HIIT sessions were carried out twice a week and their intensity was monitored daily using the Borg’s Scale (6-20) of the rate of perceived exertion (RPE) (19). Each participant was individually asked: "How did you perceive exertion during the exercise execution?". The first four weeks had an RPE of 12-13, indicating a somewhat hard level. The following four weeks were at an RPE of 15-16, signifying a hard level. In the last two weeks, the RPE was 18-19, representing an extremely hard level. However, the HR was not monitored during the intervention.”.

P10, L301-208: Also, we explicitly added the following sentence in the limitations part of the discussion section: “Secondly, it is important to acknowledge that relying solely on participants' RPE without concurrent heart rate monitoring introduces a potential limitation. While RPE is a valuable subjective measure of effort, incorporating heart rate data would have provided an additional objective parameter to quantify the actual physiological intensity experienced by participants during each session. This absence of heart rate monitoring restricts the precision of intensity characterization and may limit the generalizability of our findings to a broader context where objective intensity measures are commonly employed.”.

Discussion:

REVIEWER: Finally, I believe the discussion section needs to be reviewed carefully to condense the volume. I would suggest: (1) combining paragraphs 2 and 3 (starting with the 1st line from paragraph 3, Lines 248-249, which summarise the findings nicely), and making efforts to remove reference to the methods adopted by the comparative studies (Lines 244-245);

AUTHORS: P9, L239-251: We have changed accordingly, and now reads: “The results of the present study showed that online HIIT can be an effective stimulus for weight loss and decreasing BMI. Significant weight loss (effect size: 0.41, moderate, -3.22%) and a decrease in body mass index (effect size: 0.24, moderate, -2.72%) compared to the control group. This aligns with recent studies reporting body composition improvements within 8 weeks (8), and 15 weeks (20) of HIIT in sedentary young females. The effectiveness of online HIIT in promoting weight loss and reducing BMI in our study may be attributed to increased post-exercise energy expenditure. HIIT is known to induce excess post-exercise oxygen consumption (EPOC), leading to elevated oxygen consumption and calorie expenditure, contributing to a negative energy balance and, consequently, weight loss (21). Contrary to our findings, previous studies (22–24) reported no significant changes in body weight, BMI, and body fat percentage after HIIT in sedentary young females. Differences in participant characteristics and exercise program content (duration, frequency, intensity, volume, and progression) could contribute to these varying results.”.

REVIEWER: (2) combining paragraphs 4 and 5, to reduce the amount of repetition and again making efforts to remove reference to the methods adopted by the comparative studies.

AUTHORS: P9, L252-270: We have changed accordingly, and now reads: “A 10-week online HIIT intervention led to significant improvements in VO2 max, power output (W/kg), and power output (W) when compared to the control group. This is in line with prior research, such as Lu et al. (8), reported a substantial increase in VO2 max (+18.8%) following an 8-week HIIT in sedentary young females. Another study demonstrated noteworthy enhancements in VO2 max (approximately 22%) with cycling HIIT performed for 18–30 minutes per week over 12 weeks (22). Moreover, evidence suggests that both low-volume (1x4-min treadmill running at 85%–95% HRmax) and high-volume HIIT (4x4-min treadmill running at 85%–95% HRmax), executed thrice weekly over 6 weeks, exhibit equal effectiveness in enhancing aerobic capacity (VO2 max) among sedentary young women (15). A systematic review and meta-analysis on adults indicated that short-term HIIT (<12 weeks) and long-term HIIT (≥12 weeks) performed at least 3 times a week for 12 weeks resulted in increased VO2 max. Additionally, the increase in aerobic capacity was reported to be greater at longer training times (7). Another meta-analysis demonstrated that HIIT ranging from 2 to 8 weeks led to a substantial increase in VO2 max by 4.2–13.4% in healthy sedentary/recreationally active adults, emphasizing its positive impact on aerobic capacity (28). A recent systematic review underscores that HIIT, with a minimum of 4 weeks of exercise training (3 times per week, 18-30 minutes per session, 85-95% HRR), is effective for augmenting maximal aerobic capacity in women with a sedentary lifestyle (29).”.

Minor revisions

REVIEWER: Lines 28-31. The authors state: The repeated measures ANCOVA indicated a significant effect in the within-group analysis for weight (p=0.001; d=-0.96), demonstrating a significant decrease in the Experimental Group (EG) while the Control Group (CG) did not show a significant decrease in body mass index (p=0.15; d=-30 0.11) . Weight and BMI are different measures/variables. Please summarise the finding for each of these separately.

AUTHORS: P1, L28-31: Dear reviewer, thank you. We changed accordingly to “The repeated measures ANCOVA indicated a significant effect in the within-group analysis for weight (p=0.001; d=-0.96), and for BMI (p=0.001; d = 0.24), demonstrating a significant decrease in the Experimental Group (EG). The Control Group (CG) did not show any significant decrease in both body weight and BMI”.

REVIEWER: Lines 33-34. The authors state: A new within-group analysis with power output (W) revealed that EG significantly improved (p=0.01; d=-0.96), while CG did not show a significant improvement in power output (W/kg) (p=0.20; d=-0.11) . Again, W and W/kg are different variables and should not be compared. Each variable should be summarised separately.

AUTHORS: P1, L31-35: Dear reviewer, thank you. We changed accordingly to “Regarding maximal oxygen uptake (VO2 max) values, the EG exhibited a significant improvement (p=0.001; d=-1.07), whereas the CG did not demonstrate a significant improvement (p=0.08; d=-0.37). The EG significantly improved power output (W) (p=0.001; d = -0.50) and power output standardized by body weight (W/kg) (p=0.001; d=-0.96). The CG did not show a significant improvement in either power output (W/kg) and power output.”.

REVIEWER: Lines 160-161. The authors sate: The HIIT sessions were carried out twice a week and their intensity were monitored daily using both HR . I believe you should also include RPE on the end of this sentence.

AUTHORS: P4, L160-166: Dear reviewer, thank you. Considering another comment you have made on the Major revisions part, and my answer, this now reads: “The HIIT sessions were carried out twice a week and their intensity was monitored daily using the Borg’s Scale (6-20) of rate of perceived exertion (RPE) (19).”.

REVIEWER: Lines 184-186. The authors state: In the initial two weeks, participants engaged in 8 minutes of Tabata, gradually increasing the volume as their physical condition improved, ultimately reaching a 20-minute Tabata duration . This contradicts what the authors stated previously (Lines 156- 157): In the first month, they exercised for 20 seconds of intense intervals with 10 seconds of recovery, repeating it for 4 minutes . Please revise for clarity and provide greater detail on how the program was progressively overloaded (i.e., when and by how much did you increase your Tabata volume?).

AUTHORS: P5, L181-187: Dear reviewer, thank you. We appreciate the reviewer's astute observation, and we would like to clarify the progression of Tabata volume in our study. In the initial two weeks of the intervention, participants engaged in 8 minutes of Tabata, consistent with the protocol of 20 seconds of intense intervals with 10 seconds of recovery, repeated for 4 minutes. Following this initial phase, the volume was gradually increased in subsequent weeks. Specifically, we progressively overloaded the Tabata duration by adding an extra minute every two weeks, reaching the final duration of 20 minutes by the end of the 10-week intervention. We hope this clarifies the progression of Tabata volume throughout the study period. Given this, we changed the manuscript for better readability and clarity. Now reads: “During the first two weeks of the intervention, participants followed an 8-minute Tabata protocol, adhering to 20 seconds of intense intervals with 10 seconds of recovery, repeated for 4 minutes. Subsequently, we systematically increased the Tabata duration in the following weeks. More precisely, we implemented a progressive overload strategy by incorporating an additional minute every two weeks. By the conclusion of the 10-week intervention, the Tabata duration had reached a total of 20 minutes.”.

REVIEWER: Table 3. In the 4th exercise for the 2nd session the authors list deadweight . Could you please clarify what is meant by this exercise or provide an alternate description.

AUTHORS: P4, L173: Dear reviewer, thank you for this observation. This was a typo. We wanted to write “Deadlift w/elastic band” instead of “deadweight”. We changed accordingly.

Reviewer 3 Report (New Reviewer)

Comments and Suggestions for Authors

Good work overall on your paper, below are a number of suggestions and edits to make to your document.

Line 22 - define what is a sedentary person

line 24 and 25 remove the "n=25" since you have already stated the sample size. Also add in that these values are in mean +/- SD

In the results section don't just give the significance but the actual change scores as well

line 35 change "Load" to "load"

line 60 improve your phrasing for what you mean by "improve waist circumference" and the following metrics

line 64 clean up your grammar in this sentence

line 77 "presential approach" what is that?

Was the physical testing and training performed at the same time each day?

Was the seat height for the subjects standardized in some way or uniform?

Line 149 remove "concerning the training regimen,"

line 161 what HR monitor did you use and how did you use "both" HR. Were there two monitors?

line 166 how did you determine the max heart rate? through testing or formulas.

Table 3, make sure that you have HRmax on each line for third session

what is deadweight?

Table 2, put age and height in a separate table. 

line 218 how did you perform a separate analysis, is this different from what you denoted in the methods?

Was the VO2max tested or was it based on predicted values?

Did you measure any changes in resting BP or HR from the groups over the training?

line 257 clean up the phrasing here.

Otherwise this is great work. Keep it up. 

Comments on the Quality of English Language

Overall well written with a few different grammatical and spelling areas occasionally. 

Author Response

Reviewer 3, Minor Revisions:

AUTHORS: Dear Reviewer, thank you so much for your great and detailed review. Below are all changes made, referencing each page and the lines where the changes were made. All changes in the manuscript regarding your suggestions are highlighted in GREEN.

Minor revisions

REVIEWER: Line 22 - define what is a sedentary person.

AUTHORS: P1, L22-23: Dear reviewer, thank you. The definition of a sedentary individual in the context of sports sciences is attributed to the individual who engages in minimal physical activity, typically characterized by a lack of regular exercise or structured fitness routines. Sedentary behavior often involves prolonged periods of sitting or low-energy activities with minimal physical exertion. We have added the following: “(defined as an individual who lacks regular exercise or structured fitness routines)”.

REVIEWER: line 24 and 25 remove the "n=25" since you have already stated the sample size. Also add in that these values are in mean +/- SD

AUTHORS: P1, L24-27: Dear reviewer, thank you. We have changed accordingly. Now reads: “The experimental group (EG) with 24 women (mean±SD: age 21.21 ± 2.15 years, weight 61.16 ± 8.94 kg, height 163.96 ± 4.87 cm, body mass index [BMI] 22.69 ± 2.49 kg/m2), and the control group (CG) with another 24 women (mean±SD: age 20.50 ± 1.29 years, weight 62.0 ± 6.65 kg, height 163.92 ± 4 .89 cm, body mass index 23.04 ± 1.74 kg/m2).”.

REVIEWER: In the results section don't just give the significance but the actual change scores as well.

AUTHORS: P5-6, L210-212: Dear reviewer, thank you. Table 2 already reported the changes in percentages for each analyzed measure. If we repeat it in the table description it would be redundant.

REVIEWER: line 35 change "Load" to "load"

AUTHORS: P1, L37: Dear reviewer, thank you. We have changed accordingly.

REVIEWER: line 60 improve your phrasing for what you mean by "improve waist circumference" and the following metrics

AUTHORS: P2, L60-64: Dear reviewer, thank you. We have changed accordingly: “HIIT has gained recognition as a popular alternative exercise approach compared to traditional moderate-intensity continuous exercise. This method aims to reduce body fat percentage, lower the resting heart rate, and improve aerobic fitness in a shorter timeframe, particularly among sedentary individuals (7–9).”.

REVIEWER: line 64 clean up your grammar in this sentence

AUTHORS: P2, L64-66: Dear reviewer, thank you. We have changed accordingly: “Similarly, HIIT provides significant enhancements in VO2 max. This contributes to improvements in both aerobic and anaerobic fitness within a brief period, surpassing the benefits offered by alternative training methods. (10).”.

REVIEWER: line 77 "presential approach" what is that?

AUTHORS: P2, L77-78: Dear reviewer, thank you. Well, our HIIT intervention was conducted in an online context. This is the main characteristic of our study. So, we were stating that, usually, HIIT interventions are conducted in person. To avoid confusion, we added the following “While previous literature has predominantly delivered HIIT in person, rather than conducting them online (7)”.

REVIEWER: Was the physical testing and training performed at the same time each day?

AUTHORS: Dear reviewer, thank you. No. The pre-and post-assessments were conducted on separate days. No testing was conducted during the intervention. We strictly followed the CONSORT guidelines for RCT to ensure that all methods were completely transparent.

REVIEWER: Was the seat height for the subjects standardized in some way or uniform?

AUTHORS: Dear reviewer, thank you. We did not measure “sitting height”. We only measured the standing height. This measure was obtained using a SECA stadiometer, and the absolute values were not standardized by any other measurement. This is explicit in P3-L132-133: “was measured using a stadiometer (Type SECA 225, Hamburg, Germany) accurate to the nearest 0.1 cm.”.

REVIEWER: Line 149 remove "concerning the training regimen,"

AUTHORS: P4, L150: Dear reviewer, thank you. We removed.

REVIEWER: line 161 what HR monitor did you use and how did you use "both" HR. Were there two monitors?

AUTHORS: P4, L161-167: Dear reviewer, thank you. As this was a previous error of mine, and as it was in this part that another reviewer had some concerns, I had to change this accordingly. Now read: “The HIIT sessions were carried out twice a week and their intensity was monitored daily using the Borg’s Scale (6-20) of the rate of perceived exertion (RPE) (19). Each participant was individually asked: "How did you perceive exertion during the exercise execution?". The first four weeks had an RPE of 12-13, indicating a somewhat hard level. The following four weeks were at an RPE of 15-16, signifying a hard level. In the last two weeks, the RPE was 18-19, representing an extremely hard level. However, the HR was not monitored during the intervention.”.

REVIEWER: line 166 how did you determine the max heart rate? through testing or formulas.

AUTHORS: P3-4, L135-148: Dear reviewer, thank you. We have previously described how we obtained the HRmax: “All participants performed a sub-maximal incremental fitness test on a cycle ergometer (Viasprint 150 P cycle-ergometer) connected to Jaeger Master Screen gas analyzer. Determination of the ventilatory anaerobic threshold (VAT) was based on the respiratory gas exchange method (RER) (RER = CO2 production/O2 consumption), which detected the VAT at the point at which RER exceeds the cut-off value of 1.0 (17), as well as the obtained maximal oxygen uptake. First, an RS800CX Polar monitor (Polar Electro, Finland) was used to monitor and record maximal heart rate (HRmax) during pre- and post-assessments. In this sense, the protocol consisted of a submaximal incremental test with a fixed cadence of 60 revolutions per minute (rpm). The warm-up started at 0 W and the workload was increased by 10 W every min until min 5. The participant began the exercise phase pedaling at 50 W, and the workload was increased to 25 W every two min. After each increase, the workload remained stable for the next 2 minutes. The submaximal test ended once the VAT was reached. The highest power output (W) reached during the cycle ergometer test was registered. Finally, the highest power output reached was relativized by each participant's body weight (W/kg).”.

REVIEWER: Table 3, make sure that you have HRmax on each line for third session

AUTHORS: Dear reviewer, thank you. We have changed.

REVIEWER: what is deadweight?

AUTHORS: Dear reviewer, thank you. We have changed to deadlift w/elastic bands. It was a typo.

REVIEWER: Table 2, put age and height in a separate table. 

AUTHORS: Dear reviewer, thank you for noticing this issue. We have removed age and height, as these two measures are not controlled by any intervention.

REVIEWER: line 218 how did you perform a separate analysis, is this different from what you denoted in the methods?

AUTHORS: P6, L220: Dear reviewer, thank you. No, it is the same analysis. However, we have to perform the statistics in separate files in the Statistica software.

REVIEWER: Was the VO2max tested or was it based on predicted values?

AUTHORS: P4, L150: Dear reviewer, thank you. Thi information is outlined in P3-4, L135-148: ““All participants performed a sub-maximal incremental fitness test on a cycle ergometer (Viasprint 150 P cycle-ergometer) connected to Jaeger Master Screen gas analyzer. Determination of the ventilatory anaerobic threshold (VAT) was based on the respiratory gas exchange method (RER) (RER = CO2 production/O2 consumption), which detected the VAT at the point at which RER exceeds the cut-off value of 1.0 (17), as well as the obtained maximal oxygen uptake.”.

REVIEWER: Did you measure any changes in resting BP or HR from the groups over the training?

AUTHORS: Dear reviewer, thank you. No. It was out of the scope of this study. However, we consider that those two measures are relevant for sedentary individuals.

REVIEWER: line 257 clean up the phrasing here.

AUTHORS: P4, L150: Dear reviewer, thank you. We have changed accordingly.

This manuscript is a resubmission of an earlier submission. The following is a list of the peer review reports and author responses from that submission.

Round 1

Reviewer 1 Report

Comments and Suggestions for Authors

The manuscript has a serious methodological error. Evaluate the sample on the cycle ergometer and prescribe running and jumping training. In other words, there is no load control relationship to know the effectiveness of the training.

Evaluating on a cycle ergometer and prescribing these intensities for running is a huge methodological error.

Reviewer 2 Report

Comments and Suggestions for Authors

I have reviewed the manuscript titled "Effects of 10-week interval training on body composition and performance variables in physically inactive women" for consideration for publication in Healthcare.

It is a very interesting study that will probably encourage other authors to carry out more studies. Congratulations to the authors.

However, prior to its potential publication, there are some issues that in my opinion should be reviewed.

In the text there are several abbreviations that appear prior to their explanation or, sometimes, in duplicate. I indicate several examples that the authors should correct:

- High-intensity Level training (HIIT) is shown again on page 2 line 66, when this abbreviation had already been made on page 1 line 42.

- EG and CG appear on page 2 line 76/77 without explanation.

- HR appears on page 3 line 114 and its explanation appears a few lines later on line 127.

- VO2max appears on page 8 line 240, having been mentioned numerous times throughout the text.

Based on what the authors indicate, the reader must understand that all the subjects who participated in the study completed 100% of the sessions and that all of them completed the study (24-24). The authors should clarify if this was the case and if not, if they considered any minimum number of sessions as an exclusion criterion, since they have not indicated it. Please clarify this point so we can make a decision.

Reviewer 3 Report

Comments and Suggestions for Authors

Please refer to the attached document.

Comments on the Quality of English Language

Please refer to the attached document. 
